# Fuzzy Clustering Algorithm with Non-Neighborhood Spatial Information for Surface Roughness Measurement Based on the Reflected Aliasing Images

**DOI:** 10.3390/s19153285

**Published:** 2019-07-26

**Authors:** Hang Zhang, Jian Liu, Lin Chen, Ning Chen, Xiao Yang

**Affiliations:** State Key Laboratory of Advanced Design and Manufacture for Vehicle Body, Hunan University, Changsha 410082, China

**Keywords:** fuzzy clustering, image segmentation, spatial information, surface roughness

## Abstract

Due to the limitation of the fixed structures of neighborhood windows, the quality of spatial information obtained from the neighborhood pixels may be affected by noise. In order to compensate this drawback, a robust fuzzy c-means clustering with non-neighborhood spatial information (FCM_NNS) is presented. Through incorporating non-neighborhood spatial information, the robustness performance of the proposed FCM_NNS with respect to the noise can be significantly improved. The results indicate that FCM_NNS is very effective and robust to noisy aliasing images. Moreover, the comparison of other seven roughness indexes indicates that the proposed FCM_NNS-based *F* index can characterize the aliasing degree in the surface images and is highly correlated with surface roughness (*R*^2^ = 0.9327 for thirty grinding samples).

## 1. Introduction

Surface roughness refers to the unevenness of surface, which is an important parameter in various technological and machining systems. For example, in aeronautical engineering, the flight dynamics and the wear are affected by the surface roughness of wing aircraft [1]. In agricultural spraying, the surface roughness is used to characterize leaf surface wettability [2,3]. In textile industry, the roughness measurement of fabric surface makes the evaluation of texture properties more efficient and objective [4]. The performance of optical systems is also influenced by the surface roughness of its components [5,6]. Usually, the final process of machining is the grinding process, which can directly affect the product aesthetics and roughness level. Since the roughness of workpiece has a great influence on its contact stiffness, friction wear, corrosion resistance, and fatigue resistance [7,8], the measurement of the grinding surface roughness will be studied in this paper.

Generally, the measurement techniques of surface roughness can be divided into contact and non-contact methods [9,10]. The most common contact type is the stylus method, which has been used extensively in a lot of systems and performs well [11]. The main attribution for the wide use of stylus device is the existence of traceable standards, while the stylus method still has some drawbacks. First of all, it can’t work online and its efficiency is low. Besides, the stylus tip cannot reach into all the valleys of the surface and the diamond stylus may scratch the surface. Under this circumstance, non-contact measurement methods have gained increasing attention in recent years. The main non-contact methods include optical systems [12,13], vibration signals analysis [14,15], and machine vision techniques [16,17]. Among these methods, the machine vision-based measurement is very efficient, flexible, and cost-effective. In addition, the machine vision methods can support online measurement.

In the process of machine vision measurement, the images of different roughness samples are firstly collected by an industrial camera. Then the roughness correlated features are extracted from these images to establish a roughness prediction model. Finally, the roughness of the unknown surface can be measured using these image features and the trained prediction model. In the conventional machine vision measurement methods, the roughness correlated features are usually extracted from the gray scale images [10,18,19]. Wei et al. presented a Gray Level Co-occurrence Matrix and Support Vector Machine-based method. This method was applied on the roughness measurement of a hole and achieved high predictive accuracy [10]. B. Ramamoorthy et al. presented a frequency domain features and neural network based surface roughness measurement. Experiment results showed that this intelligent visual measurement could obtain high measurement accuracy [20]. However, the sensitivity of these features to the roughness parameter may be affected, because the gray scale image lost the color information and the features mentioned above are all extracted from these degraded images. To address this problem, the color information has been studied to analyze the machined surface images [7,17,21]. Liu et al. proposed a color difference index to measure the roughness of grinding surface. This color index performs well on the surface roughness measurement, and presented high level of robustness [7].

As color images have more detail information, more accurate features can be extracted from these images to identify targets [22,23], but in previous works [7,17,21], only simple color indexes are proposed to characterize the surface images, which ignore the correlation between pixels and the spatial information of pixels. Moreover, appropriate image processing algorithm can be applied to fully characterizing the surface images and accurately measuring the surface roughness [11,24,25]. In this study, fuzzy c-means (FCM) clustering is employed to further analyze the color images reflected by grinding surface. Clustering [26,27,28] is the process of distinguishing and classifying things in accordance with some certain rules. FCM is the widely used clustering methods, it is based on fuzzy set theory [29] and follows an iterative process to obtain clusters and fuzzy cluster memberships [30,31]. Compared with the hard clustering scheme, in which each pixel of the image only can be classified to one cluster, the FCM algorithm allows each pixel to belong into all clusters with meaningful degree of membership [32,33]. Therefore, the FCM algorithm can preserve more image information and present more robust characteristics for ambiguity [34].

The research in our previous work [7,17] indicates that the aliasing degree of color images will strengthen as the surface roughness increases. In order to measure the surface roughness, the image aliasing degree can be analyzed by the FCM algorithm firstly. However, the color images reflected by grinding surface have high noise levels due to the effect of surface textures and other machining marks. Because the FCM algorithm is sensitive to noise and outliers [35,36], the direct use of this algorithm may not provide a good roughness measurement accuracy.

The spatial information have been introduced to enhance the robustness of original FCM algorithms [35,36,37]. Ahmed et al. [37] proposed FCM_S by introducing neighborhood spatial information. The intensity inhomogeneity of magnetic resonance images can be compensated by this algorithm. To reduce the computation cost of FCM_S, Cai et al. [38] presented a fast generalized FCM (FGFCM) algorithm by introducing a similarity measure. The computation cost of FGFCM is very little and the performance is well enhanced. However, some parameters are needed in these algorithms to control the balance among noise and image details. To overcome this problem, Stelios et al. [35] proposed a robust fuzzy local information C-Means clustering algorithm (FLICM). There is no parameter selection in this algorithm. But when the image is seriously corrupted, the above spatial information cannot achieve satisfactory results [39,40]. Thus, Zhao [40] proposed a novel FCM algorithm by incorporating non-local spatial information. The experiment results indicate that this algorithm performs well than the local spatial information based FCM algorithms. However, due to the fixed structures of neighborhood window, the quality of spatial information obtained from the neighborhood pixels may be affected by the noise. To augment the spatial sampling locations in the convolutional neural networks (CNNs), Dai et al. [41] proposed a deformable convolutional networks. Inspired by this idea, a robust FCM algorithm with non-neighborhood spatial information (FCM_NNS) is presented. The results indicate that FCM_NNS is effective and robust to noisy aliasing images.

The rest of this paper is organized as follows. The spatial information based FCM algorithms are briefly described in Section 2. Section 3 introduces our motivation and the FCM_NNS algorithm. The results are analyzed in Section 4. Section 5 is the conclusions.

## 2. Preliminary Theory

### 2.1. Fuzzy C-Means (FCM) Algorithm

The FCM algorithm was proposed by Dunn [42]. Subsequently, Bezdek [33] extended the objective function to a more general form. The FCM algorithm follows an iterative process to obtain clusters and fuzzy cluster memberships by minimizing the objective function:(1)$$J_{m}=\sum_{i=1}^{c}\sum_{j=1}^{N}u_{ij}^{m}{\left\|x_{j}-v_{i}\right\|}^{2}$$
where m∈(1,∞) is a weighting exponent, $X=\{x_{1},x_{2},\ldots ,x_{N}\}\subseteq R^{n}$ is the dataset in the n-dimensional vector space, ‖·‖ denotes the Euclidean norm. $v_{i}\left(i=1,2,\ldots ,c\right)$ is the cluster center of *i*th cluster, *N* and *c* are the number of input data and clusters, respectively. The array $U={\{u_{ij}\}}_{c\times N}$ represents a membership matrix satisfying:(2)$$U\subseteq \{u_{ij}\in \left[0,1\right]|\sum_{i=1}^{c}u_{ij}=1,\forall j\;\mathrm{and}\;0<\sum_{j=1}^{N}u_{ij}<N,\forall i\}$$

Then, a solution can be obtained by updating the membership $\{u_{ij}\}$ and clusters $\{v_{i}\}$ as follows:(3)$$u_{ij}=\frac{1}{\sum_{r=1}^{c}{\left(\frac{\left\|x_{j}-v_{i}\right\|}{\left\|x_{j}-v_{r}\right\|}\right)}^{\frac{2}{m-1}}}$$
(4)$$v_{i}=\frac{\sum_{j=1}^{N}u_{ij}^{m}x_{j}}{\sum_{j=1}^{N}u_{ij}^{m}}$$

If $\max\left\|V^{\left(k+1\right)}-V^{\left(k\right)}\right\|<\varepsilon$ or the number of iterations k>T then stop, the parameters ε  and T are iterative termination conditions.

### 2.2. Fuzzy Clustering with Constraints (FCM_S) and Its Variants

Ahmed et al. [37] presented a FCM_S, which allows the central pixel to be influenced by its neighborhood. The expression of objective function is as follows:(5)$$J_{FCM\_S}=\sum_{i=1}^{c}\sum_{j=1}^{N}u_{ij}^{m}{\left\|x_{j}-v_{i}\right\|}^{2}+\frac{\alpha }{N_{R}}\sum_{i=1}^{c}\sum_{j=1}^{N}u_{ij}^{m}{\sum_{r\in N_{j}}\left\|x_{r}-v_{i}\right\|}^{2}$$
where $x_{r}$ denotes the neighbor of $x_{j}$, and $N_{j}$ denotes the pixels within neighborhood window around $x_{j}$. α is the neighborhood term parameter, and $N_{R}$ is the cardinality. In addition, the objective function satisfies the constraints shown in Equation (2).

Chen et al. [43] presented FCM_S1 and FCM_S2 algorithms by adopting the mean and median values of the pixels within neighborhood window. Szilagyi et al. [44] proposed an enhanced FCM (EnFCM) to further reduce the computation cost. In addition, Cai et al. [38] presented a fast generalized FCM (FGFCM). This algorithm defines a new similarity measure. Its computation time is similar to EnFCM.

### 2.3. Fuzzy Local Information C-Means (FLICM) Clustering and Its Variants

The above FCM algorithms all need some parameters to control the balance among noise and image details. To compensate this drawback, Stelios et al. [35] proposed a fuzzy local information c-means (FLICM). There is no parameter selection in this algorithm. Specifically, FLICM introduces a fuzzy factor $G_{ij}$ defined as:(6)$$G_{ij}=\sum_{\begin{array}{c} r\in N_{j}\\ r\neq j \end{array}}\frac{1}{d_{jr}+1}{\left(1-u_{ir}\right)}^{m}{\left\|x_{r}-v_{i}\right\|}^{2}$$
where the *j*th is the central pixel in the neighborhood window, $N_{j}$ is the pixels around the *j*th pixel, $d_{jr}$ represents the distance between pixels j and r.

By incorporating the fuzzy factor $G_{ij}$, the objective function of FLICM is expressed as follows
(7)$$J_{FLICM}=\sum_{i=1}^{c}\sum_{j=1}^{N}\left[u_{ij}^{m}{\left\|x_{j}-v_{i}\right\|}^{2}+G_{ij}\right]$$

More recently, Gong et al. [36,45] presented two variants of FLICM algorithm: fuzzy local information c-means with trade-off weighted fuzzy factor and kernel method (KWFLICM) and reformulated fuzzy local information c-means (RFLICM). The results show that the new algorithms are more effective.

### 2.4. Fuzzy Clustering Algorithm with Non-Local Spatial Constraint and Its Variants

Zhao [40] proposed a FCM algorithm by incorporating non-local spatial information (FCM_SNLS). When the image is seriously corrupted, the FCM_SNLS is more robust and effective than the local spatial information based FCM algorithms. By using the definition of neighborhood configuration, the non-local spatial information can be calculated as follows:(8)$$\vartheta_{j}=\sum_{p\in W_{j}^{r}}w_{jp}x_{p}$$
where $W_{j}^{r}$ is a r×r search window around the *j*th pixel, the weight $w_{jp}$ satisfies $0\leq w_{jp}\leq 1$ and $\sum_{p\in W_{j}^{r}}w_{jp}=1$. Specifically, the $w_{jp}$ is defined as follows:(9)$$w_{jp}=\frac{1}{Z_{j}}\exp\left(-{\left\|x\left(N_{j}\right)-x\left(N_{p}\right)\right\|}_{2,\sigma }^{2}/h^{2}\right)$$
where $x\left(N_{j}\right)$ is a gray level vector within a s×s square neighborhood $N_{j}$ centered at the *j*th pixel. ${\left\|x\left(N_{j}\right)-x\left(N_{p}\right)\right\|}_{2,\sigma }^{2}$ represents a Gaussian weighted Euclidean distance, σ>0 is the standard deviation of the Gaussian kernel. The parameter h is the filtering degree, and $Z_{j}$ is a constant.

More recently, Shang et al. [46] proposed a clone kernel spatial FCM (CKS_FCM). CKS_FCM improves the robustness to noise by incorporating both local and non-local spatial information.

## 3. Fuzzy C-Means Clustering Algorithm with Non-Neighborhood Spatial Information

The spatial information mentioned above are obtained from the pixels within neighborhood window, which may be affected by the noise due to the fixed structures of neighborhood window. The non-neighbor spatial information has not been considered yet. In this section, a robust FCM algorithm with non-neighbor spatial information (FCM_NNS) is proposed, and the aliasing degree of the surface images is analyzed by the proposed FCM_NNS. Notations and their descriptions used in the proposed algorithm are shown in Table 1.

### 3.1. Motivation and Notation

According to the analysis in reference [7], it can be known that the reflected area of the red and green points vary due to the different surface roughness. If the roughness increases, the reflected area will be larger and the aliasing area of two points will also be larger. The virtual images of the red and green points reflected by the surface with different roughness are illustrated in Figure 1.

In the process of measurement, the red and green points in Figure 1 are designed as a block, as shown in Figure 2a. From the above analysis, it can be found that the aliasing area increases monotonically as the roughness increases. Based on this theory, the feature index correlated with roughness can be designed by evaluating the aliasing effect. However, in the previous work, researches only count the number of pixels with the same red and green brightness values [21], or calculate the absolute difference between the brightness values of red and green components [7] to characterize the aliasing images. These approaches are obviously not comprehensive and reasonable, since the correlation between pixels and the spatial information of pixels are ignored. In this study, FCM algorithm is employed to analyze the aliasing images. The FCM is suitable for describing the uncertainty of the aliasing image, as shown in Figure 2b, and extracting more appropriate image feature index. However, the segmentation performance of conventional FCM will be influenced when dealing with the high noisy aliasing images. Thus, a robust FCM algorithm with non-neighbor spatial information (FCM_NNS) is presented to address this problem.

### 3.2. Initializing Cluster Centers

The conventional FCM clustering is a local search algorithm, which is sensitive to the initial cluster centers. If the initial cluster centers are appropriate, the convergence of the algorithm will be fast. Thus, it is crucial to initialize appropriate cluster centers.

When measuring the surface roughness by the red and green color blocks, the aliasing degree should be evaluated to characterize the surface image. It is obvious that the image is divided into three regions: red, green and aliasing, as shown in the Figure 2b, so the number of cluster centers can be set to 3. In [21], Liu et al. proposed color distribution statistical matrices (CDSM) to characterize the aliasing effect. The CDSM is a two-dimensional matrix to gather statistics of the red and green brightness levels. It represents the number of pixels with any red and green brightness level, as illustrated in Figure 3a. Then, the sum of the data in the diagonals of the CDSM is considered to the aliasing region and used to measure the surface roughness. It is not that reasonable to measure surface roughness in this way, but it can provide inspiration to initialize the cluster centers.

In the aliasing images, the brightness value of the green component decreases gradually from left to right, while the value of the red component increased from left to right, as illustrated in Figure 3b. So in the middle of the image, there is an aliasing region whose absolute value between the red and green brightness values is small. On both sides of aliasing region, there are two pure color regions with a large color difference. Therefore, if we set a threshold, all the pixels in the image can be preliminarily divided into three regions according to the color difference, as shown in Figure 3c. Specifically, Figure 3c is the CDSM of the image shown in Figure 3b, the blue points represent the pixels. If the threshold is set to be 10, the pixels in this image can be preliminarily segmented into red, aliasing and green regions. Then, the average brightness values of all the pixels in each region can be obtained as initial cluster centers. In summary, the process of initializing cluster centers is shown in Algorithm 1.

**Algorithm 1** The process for initializing cluster centers**Input**: aliasing image *I*;**Output:** initial cluster centers *V*_1_;**Begin**1. Calculate the color difference between the red and green component of each pixel in the image;2. Set a threshold *K*, and divide pixels into three regions according to the color difference;3. Calculate the average red and green brightness values of all the pixels in each region to obtain the initial cluster centers *V*_1_.**End**

### 3.3. Image Filtering Based on Non-Neighborhood Spatial Information

To improve the segmentation performance, some new FCM algorithms have been presented by introducing spatial information. However, when dealing with aliasing images, these algorithms often generate unsatisfactory results. Thus, a robust FCM by incorporating non-neighborhood spatial information is proposed here.

As shown in Figure 4a, in the direction of x, the green component of the aliasing image presents a linear decrease tendency. And in the direction of y, each column pixels basically has the same brightness value. Thus, this non-neighborhood spatial information in two directions can be used to improve the segmentation performance. It is not difficult to find that the non-neighborhood spatial information used in this task is to set the size of the window containing spatial information to an extreme case, which can make full use of the prior knowledge in the aliasing image. In addition, the neighborhood spatial information needs to calculate the filtering value of central pixel window by window, while the proposed non-neighborhood spatial information can directly obtain the filtering values of all the pixels in the non-neighborhood window, which greatly improves the efficiency of FCM algorithm. As shown in Figure 4a, the neighborhood spatial information needs to calculate 25 times through window movement to obtain the filtering values of all the pixels, while the proposed non-neighborhood spatial information only need five times.

For a pixel $x_{i,j}$ in the aliasing image, the non-neighborhood spatial information in two directions provide two reference brightness values, which can be obtained by fitting the brightness value of *i*th row pixels and averaging the brightness values of *j*th column pixels respectively, as shown in Figure 4a. The calculation formula for the two reference values $x_{p1},x_{p2}$ are expressed as follows:(10)$$x_{p1}=b_{i}\times j+a_{i},\;\;\;i=1,\ldots ,t,\;\;j=1,\ldots ,r$$
(11)$$x_{p2}=\frac{1}{t}\sum_{k=1}^{t}x_{k,j}\;,\;\;\;j=1,\ldots ,r$$
where $x_{i,j}$ denotes the pixel in *i*th row, *j*th column. The parameters t,r is the size of non-neighborhood window, and the linear regressive coefficient *a*, *b* can be obtained as follows:(12)$$b_{i}=\frac{\sum_{k=1}^{r}\left(k-\bar{x}\right)\left(x_{i,k}-\bar{y}\right)}{\sum_{k=1}^{r}{\left(k-\bar{x}\right)}^{2}}=\frac{12\left(\sum_{k=1}^{r}kx_{i,k}-\frac{r+1}{2}\sum_{k=1}^{r}x_{i,k}\right)}{\left(r-1\right)r\left(r+1\right)}\;,\;\;i=1,\ldots ,t$$
(13)$$a_{i}=\bar{y}-b\bar{x}=\frac{1}{r}\sum_{k=1}^{r}x_{i,k}-b\times \frac{1}{r}\sum_{k=1}^{r}k=\frac{1}{r}\sum_{k=1}^{r}x_{i,k}-\frac{\left(r+1\right)b}{2}\;\;,\;\;\;\;\;\;i=1,\ldots ,t$$
where $\bar{x}$ is the average of column numbers, and $\bar{y}$ is the average brightness values of pixels in *i*th row.

Then, by incorporating non-neighbor spatial information, a new image η can be generated as follows:(14)$$\eta_{i,j}=\frac{x_{i,j}+\sum_{k=1}^{2}E_{k}x_{pk}}{1+\sum_{k=1}^{2}E_{k}}$$
where $x_{p1},x_{p2}$ are the reference values of pixel $x_{i,j}$, $E_{1},E_{2}$ are the similarity measure which are expressed as follows:(15)$$E_{k}=\frac{{\left\|x_{i,j}-x_{pk}\right\|}^{2}}{\lambda_{k}\delta_{k}^{2}},\;\;\;k=1,2$$
where $\lambda_{1},\lambda_{2}$ are two scale factors playing a role similar to factor $\lambda_{s},\lambda_{g}$ in FGFCM, and $\delta_{1},\delta_{2}$ are defined as:(16)$$\delta_{1}=\sqrt{\frac{\sum_{k=1}^{r}{\left\|x_{i,k,p1}-x_{i,k}\right\|}^{2}}{r}}$$
(17)$$\delta_{2}=\sqrt{\frac{\sum_{k=1}^{t}{\left\|x_{k,j,p2}-x_{k,j}\right\|}^{2}}{t}}$$
where $x_{i,j}$ represents the brightness value of the pixel in the *i*th row, *j*th column, and $x_{i,j,p1},x_{i,j,p2}$ represents its reference brightness values.

There are many noise points in the aliasing images, which will damage the segmentation performance of the conventional FCM. To address this problem, the brightness values are adjusted according to the distance between the brightness value of each pixel and its reference value. As shown in Figure 4c, if the distance is large, the possibility that the pixel is a noisy point will be high, and the adjustment of the brightness value will also be large. Whereas the adjustment will be small when the distance is small. So the new image obtained by incorporating non-neighborhood spatial information can not only preserve robustness and noise insensitiveness, but also preserve details in aliasing images. In addition, the red and green components should be dealt through the above method respectively. Thus, the image segmentation results can be improved.

The non-neighborhood spatial information can be calculated by Equations (10)–(17), which is to set the size of the neighborhood window to an extreme case. It should be noted that the scale factors $\lambda_{1},\lambda_{2}$ in Equation (15) have heavily influence on the effectiveness of the non-neighbor spatial information. Specifically, too big values of $\lambda_{1},\lambda_{2}$ will cause this spatial information losing the image detail information. And too small values of $\lambda_{1},\lambda_{2}$ will lead the non-neighborhood spatial information to be still affected by the noise. Therefore, the parameters $\lambda_{1},\lambda_{2}$ should be obtained adaptively based on the noise level of the aliasing images. The calculations of adaptive $\lambda_{1},\lambda_{2}$ values of each row and column pixels are given as follows:(18)λ1,i=(1r−1∑j=1r(di,j,1−d¯i)2)12   ,   i=1,2,…,t
(19)λ2,j=(1t−1∑i=1t(di,j,2−d¯j)2)12 , j=1,2,…,r
where:(20)$$d_{i,j,k}=\left\|x_{i,j,pk}-x_{i,j}\right\|,\;\;\;i=1,2,\ldots ,t,\;\;\;j=1,2,\ldots ,r,\;\;k=1,2$$
and:(21)$${\bar{d}}_{i}=\frac{1}{r}\sum_{j=1}^{r}d_{i,j,1}\;,\;i=1,2,\ldots ,t$$
(22)$${\bar{d}}_{j}=\frac{1}{t}\sum_{i=1}^{t}d_{i,j,2}\;\;,\;\;\;j=1,2,\ldots ,r$$

### 3.4. General Framework of FCM_NNS Iteration

By incorporating the non-neighborhood spatial information term and between-cluster variation term, a robust FCM algorithm, named FCM with non-neighborhood spatial information (FCM_NNS), is proposed. The objective function of FCM_NNS is expressed as follows:(23)$$J_{m}=\sum_{i=1}^{c}\sum_{j=1}^{N}u_{ij}^{m}{\left\|x_{j}-v_{i}\right\|}^{2}+\alpha \sum_{i=1}^{c}\sum_{j=1}^{N}u_{ij}^{m}{\left\|\eta_{j}-v_{i}\right\|}^{2}-n\left(k\right)\sum_{i=1}^{c}\sum_{j=1}^{N}u_{ij}^{m}{\left\|\bar{x}-v_{i}\right\|}^{2}$$
with the following constraints:(24)$$\sum_{i=1}^{c}u_{ij}=1,\;\;u_{ij}\in \left[0,1\right],\;\;0\leq \sum_{i=1}^{N}u_{ij}\leq N$$
where *N* is the number of pixels, c is the number of clusters, $v_{i}$ denotes the center of *i*th cluster, and $u_{ij}$ is the membership of $x_{j}$ belonging to the cluster i. m is a weighting exponent and $x_{j}$ denotes the pixels of the original image. The non-neighborhood spatial information $\eta_{j}$ is the new brightness value which is adjusted by the reference values. Furthermore, α controls the effect of the non-neighborhood spatial constrain term. ${\left\|\bar{x}-v_{i}\right\|}^{2}$ is the between-cluster term and $\bar{x}$ denotes the mean of all pixels of the original image. n(k) controls the effect of the between-cluster separation term and is calculated as:(25)$$n\left(k\right)=\frac{\left(\beta /4\right)\min_{K\neq k}{\left\|v\left(k\right)-v\left(K\right)\right\|}^{2}}{\max_{j}{\left\|v_{j}-\bar{x}\right\|}^{2}}$$
where β controls the effect of the between-cluster separation term.

By minimizing Equation (23), $u_{ij}$ and $v_{i}$ can be calculated by the following update equations:(26)$$u_{ij}=\frac{1}{\sum_{l=1}^{c}{\left(\frac{{\left\|x_{j}-v_{i}\right\|}^{2}+\alpha {\left\|\eta_{j}-v_{i}\right\|}^{2}-n\left(k\right){\left\|\bar{x}-v_{i}\right\|}^{2}}{{\left\|x_{j}-v_{l}\right\|}^{2}+\alpha {\left\|\eta_{j}-v_{l}\right\|}^{2}-n\left(k\right){\left\|\bar{x}-v_{l}\right\|}^{2}}\right)}^{1/\left(m-1\right)}}$$
(27)$$v_{i}=\frac{\sum_{j=1}^{N}u_{ij}^{m}\left(x_{j}+\beta \eta_{j}-n\left(k\right)\bar{x}\right)}{\sum_{j=1}^{N}\left(1+\beta -n\left(k\right)\right)u_{ij}^{m}}$$

The details of FCM_NNS are illustrated in Algorithm 2.

### 3.5. Index Designing Based on the Clustering Results

When the results obtained by the FCM_NNS algorithm converged, a defuzzification process is applied to convert the membership partition U to a crisp partition. The common defuzzification method classifies the pixel i to the class C with the highest membership:(28)$$C_{i}=\arg_{i}\{\max\{u_{ij}\}\},\;\;\;i=1,2,\ldots ,c,\;j=1,2,\ldots ,N$$

The obtained membership matrix is converted to the segmented image by adopting Equation (28) firstly. After that, the surface image could be characterized by counting the number of pixels belonged to the aliasing class. However, the membership information in the fuzzy image is ignored in this way. It has great effects on the accurate measurement of surface roughness. To fully utilize the fuzzy information in clustering results, the index correlated with roughness can be designed as follows:(29)$$F=\sum_{i=1}^{N}z_{i}{}^{p}$$
where N is the number of pixels, $z_{i}$ is the membership degree of *i*th pixel belonging to the aliasing region class, the parameter p is similar to the fuzzification factor m, which is a weighting exponent.

The details of FCM_NNS based roughness measurement method are shown in Algorithm 3.

**Algorithm 2** The process of FCM_NNS**Input**: The original image *I*;**Output**: Cluster center *V*^(t+1)^, membership degree *U*^(t+1)^;**Begin**1. Input the image *I*, then set the number of clusters c, the fuzzification parameter m, the stopping condition ε and the maximum iteration number T, the spatial parameter α, the between-cluster parameter β, and the size of non-neighborhood search window *r* and *t*;2. Initialize the cluster centers V(1)=[v1(1),v2(1),v3(1)] using Algorithm 1;3. Obtain the non-neighborhood spatial information of each pixel using Equations (10)–(22);4. Set the iterative index t=1;5.  **While** (t<T)6.   Compute the membership functions uij(t+1) using Equation (26);7.   Compute the cluster centers vi(t+1) using Equation (27);8.   If ‖V(t+1)−V(t)‖<ε, then stop and output the membership degree *U*^(t+1)^ and cluster center *V*^(t+1)^; Otherwise, set t=t+1 and go to the 6th step.9.  **End while****End**

**Algorithm 3** The process of roughness measurement method based on FCM_NNS**Input**: The original image *I*;**Output**: Roughness related feature indexes *C* and *F*;**Begin**1. Input the image *I*;2. Obtain the cluster center *V*^(t+1)^, and the membership degree *U*^(t+1)^ using Algorithm 2;3. Obtain segmented image *I*_s_ from membership degree *U*^(t+1)^ using Equation (28).4. Compute the *C* index by counting the number of pixels in the aliasing region class of segmented image *I*_s_.5. Compute the *F* index using Equation (29)**End**

## 4. Experimental Results and Discussion

The results on one synthetic image and thirty aliasing images are discussed in this section. This study compares the performance and the efficiency of FCM_NNS with seven algorithms, such as FCM_S1 [43], FCM_S2 [43], FGFCM [38], FLICM [35], KWFLICM [36], FCM_NLS [40] and CKS_FCM [46]. Then, the effectiveness of the proposed *C* index and *F* index are validated by the comparisons between nine roughness indexes.

### 4.1. Experimental Equipment and Imaging Results

#### 4.1.1. Sample Preparation

In this experiment, for the preparation of the following imaging experiment, the surface grinder KGS-250AH is used to process thirty 40 × 60 mm^2^ grinding samples. The material of these samples is 45# steel. The roughness range is 0.0675~0.5111 μm. Six different positions on the grinding samples are measured adopting a Form Talysurf PGI 800 stylus instrument (Taylor Hobson, Leicester, UK). The results are shown in Table A1 in the Appendix A.

#### 4.1.2. Experimental Equipment

The imaging equipments are shown in Figure 5. The components of this optical system are listed as follows: (1) controller and light source; (2) a color CCD industrial camera; (3) an optical platform; (4) a red and green block; (5) a computer. During the imaging experiment, the measured surface and the color block puts 90° angle and 45° with the workbench, respectively. The axis of CCD camera is parallel to the color block, and the positions of these components remain unchanged.

#### 4.1.3. Imaging Results

The imaging results are illustrated in Figure 6. It can be found that the aliasing regions appear in the middle of the surface images, and the image aliasing degree strengthens as the roughness increases, but due to the influence of surface textures and other machining marks, the aliasing images have a high noise level. In general, when the surface roughness is larger, the noise level of the aliasing image will be higher. Thus, a robust FCM algorithm is proposed to analyze this aliasing effect. Based on the clustering results, we design an index which is significantly correlated with the surface roughness.

### 4.2. Evaluation Indexes

In this section, the clustering results and the designed roughness index are investigated. Firstly, the segmentation accuracy (*SA*) values of eight fuzzy clustering algorithms are compared. The *SA* is expressed as follows [36]:(30)$$SA=\sum_{i=1}^{c}\frac{A_{i}\cap C_{i}}{\sum_{j=1}^{c}C_{j}}$$
where c is the number of clusters, $A_{i}$ denotes the set of pixels classified to the ith class, while $C_{i}$ represents the corresponding correct classification in the reference segmented image.

However, when dealing with the aliasing images, the segmentation accuracy can’t be calculated by *SA* without ground truth. In this case, Partition coefficient $v_{pc}$ and partition entropy $v_{pe}$ [47] are employed to test the clustering performance. They are defined as follows:(31)$$v_{pc}=\frac{\sum_{j}^{N}\sum_{i}^{c}u_{ij}^{2}}{N}$$
and:(32)$$v_{pe}=\frac{-\sum_{j}^{N}\sum_{i}^{c}\left[u_{ij}\log u_{ij}\right]}{N}$$

The best performance is obtained when the value $v_{pc}$ is maximal or $v_{pe}$ is minimal. Finally, to contrast the performance of different roughness indexes, the coefficient of determination *R*^2^ [9] are adopted:(33)$$R^{2}=1-\frac{\sum {\left(y-y*\right)}^{2}}{\sum {\left(y-\bar{y}\right)}^{2}}$$
where *y* is the value measured by stylus, $\bar{y}$ is the average value of *y*, and y∗ is the predicted regression value.

### 4.3. Performance Comparison of Clustering Algorithms and Parameter Analysis

The comparative clustering algorithms and the corresponding parameters are illustrated in Table 2. For all the algorithms, the threshold ε and the maximal iteration T are set to be 10^−5^ and 300, respectively. The fuzziness index m is set to be 2. The size of the neighborhood window is set to be 3×3. In addition, we set β=6 for FCM_S1, FCM_S2, SNIS-FCM and CKS-FCM. The parameters $\lambda_{g}$ and $\lambda_{s}$ of FGFCM are set to be 6 and 3, respectively [38]. The parameters r and s are set to be 5 and 5 for FCM_NLS. Finally, the parameters α, β, $\lambda_{1}$ and $\lambda_{2}$ of the proposed FCM_NNS are analyzed in this section.

#### 4.3.1. Parameter Analysis

In this experiment, thirty aliasing images and one synthetic image are used to test the segmentation performance of eight FCM algorithms.

A method for initializing cluster centers is given in this paper. The aliasing image in Figure 6d is used for testing. The objective function values of FCM_NNS of 10 runs through random initialization are shown in Figure 7. Moreover, that value through the proposed initialization method under the threshold *K* value varying from 5 to 15 with steps of 5 are also presented in Figure 7. It can be found that the initial value obtained by the proposed method under different thresholds makes the clustering algorithm converge rapidly. And the objective function converge completely only after five iterations. However, the random initialization needs fifteen iterations to complete convergence, which proves the effectiveness of the proposed initialization method.

The size of non-neighborhood search window *r* and *t* is investigated here. The aliasing image in Figure 6d is used for testing. The row size *r* is set to vary from 10 to 400 and the column size *t* is set to vary from 10 to 600. The average *v_pc_* and *v_pe_* of 10 runs through FCM_NNS with different size of non-neighborhood search window are shown in Figure 8. It can be seen that the size of non-neighborhood search window affect the results. It is shown in Figure 8a that with each *t*, the partition coefficient *v_pc_* is low when the value of row size *r* is 10. When *r* is larger than 10, the *v_pc_* becomes higher. The curve of partition coefficient *v_pc_* rises acutely with the increase of *r* from 10 to 50. And when *r* is 400, the *v_pc_* reaches maximum value. Moreover, with each *r*, the partition coefficient *v_pc_* presents increasing tendency with *t* from 10 to 600. The partition entropy *v_pe_* presents decreasing tendency with *t* from 10 to 600. In addition, as can be known from the calculation of non-neighborhood spatial information, when the size of non-neighborhood search window is larger, the computational cost will be smaller. So, with considering the performance and computational cost, the row and column sizes are chose to be *r* = 400 and *t* = 600, respectively.

As mentioned in the third section, the scale factors $\lambda_{1}$ and $\lambda_{2}$ have great effect on FCM_NNS. Here, this study compares the segmentation performance of FCM_NNS under the parameters $\lambda_{1}$ and $\lambda_{2}$ varying from 1 to 9 in steps of 2. The aliasing image in Figure 6d is used for testing. The average $v_{pc}$ and $v_{pe}$ values of 10 runs under different scale factors are shown in Figure 9. To prove the superiority of adaptive $\lambda_{1}$ and $\lambda_{2}$ values obtained by Equations (18)–(22), the average $v_{pc}$ and $v_{pe}$ values under adaptive $\lambda_{1}$ and $\lambda_{2}$ values are also presented in Figure 9. It can be seen that the scale factors $\lambda_{1}$ and $\lambda_{2}$ affect the segmentation results. For each $\lambda_{1}$, the curve of partition coefficient $v_{pc}$ reduces acutely with the increase of $\lambda_{2}$ from 1 to 9. The curve of partition entropy $v_{pe}$ rises acutely with the increase of $\lambda_{2}$ from 1 to 9. Moreover, with each $\lambda_{2}$, the partition coefficient $v_{pc}$ reduces acutely with the increase of $\lambda_{1}$ from 1 to 9. The partition entropy $v_{pe}$ rises acutely with the increase of $\lambda_{1}$ from 1 to 9. Therefore, it is important to set an appropriate $\lambda_{1}$ and $\lambda_{2}$ values for FCM_NNS. Figure 9 shows that the adaptive values of $\lambda_{1}$ and $\lambda_{2}$ achieve the best performance, the maximum value of $v_{pc}$ and the minimum value of $v_{pe}$ are reached, at which point the values are 0.7881 and 0.3793, respectively.

In the proposed FCM_NNS, two free parameters α and β need to be analyzed. Parameter α controls the effect of the non-neighborhood filtering term and β controls the effect of the between-cluster separation term. These two parameters are studied on the aliasing image illustrated in Figure 6d. The partition coefficient $v_{pc}$ and partition entropy $v_{pe}$ of FCM_NNS with different values of α and β are shown in Figure 10.

As shown in Figure 10a, with the increase of α from 1 to 200 and of β from 1 to 8, the partition coefficient $v_{pc}$ value increases. When α is 200 and β is 8, the value of partition coefficient $v_{pc}$ reaches its maximum value, and when α is larger than 200, the value of partition entropy $v_{pc}$ begins to reduce. From Figure 10b, it can be found that the value of partition entropy $v_{pe}$ decreases significantly with the increase of α from 1 to 200. And when α is larger than 200, the value of partition entropy $v_{pe}$ begins to increase. The best performance is obtained when the $v_{pc}$ is maximal or $v_{pe}$ is minimal. Hence, the parameters setting are α=200 and β=8.

#### 4.3.2. Results on Synthetic Images

Firstly, these FCM algorithms are tested by one synthetic test image. The synthetic image is similar to the color block used in the surface imaging experiments. This image with 128×128 pixels includes two classes with 20 and 120 gray values, as illustrated in Figure 11a. The number of clusters is 2, and the test synthetic image is corrupted by Gaussian noise. Furthermore, the size of the non-neighborhood window is set to be the image size, the parameters α, β of FCM_NNS are set to be 200 and 8, and the scale factors $\lambda_{1}$ and $\lambda_{2}$ are obtained by Equations (18)–(22).

Figure 11 shows the segmentation results of a corrupted image by Gaussian noise (20%). As shown in Figure 11c–e, FCM_S1, FCM_S2 and FGFCM are seriously affected by the noise, which shows that these clustering algorithms lack enough robustness with respect to the Gaussian noise. Moreover, Figure 11f–h illustrates that FLICM, KWFLICM and FCM_SNLS can remove the majority of the noise, but the clustering results are still not satisfactory enough. In addition, Figure 11i,j shows that CKS_FCM and the proposed algorithm can remove almost all the noise and achieve satisfactory and robust results, but CKS_FCM is very time-consuming compared with the proposed algorithm.

Table 3 gives the average *SA* value of the above FCM algorithms on the noisy images corrupted by varying degrees of noise. Each experimental result is obtained by computing the mean value of 10 independent runs. It can be found out that the proposed FCM_NNS algorithm can achieve the best denoising performance compared with the other seven compared algorithms, and can get clear segmented area with high veracity as well.

#### 4.3.3. Results on Aliasing Images

Figure 12a–h and Figure 13a–h show the segmentation results on aliasing images with different roughness obtained by FCM_S1, FCM_S2, FGFCM, FLICM, KWFLICM, FCM_SNLS, CKS_FCM and FCM_NNS, respectively.

The original images are shown in Figure 6 and they were segmented into three classes corresponding to the green region, aliasing region and red region. As shown in Figure 12, it can be observed that when dealing with the aliasing image reflected by small roughness surface, all the above mentioned algorithms can obtain relatively satisfactory results, but the last algorithm has obvious advantages in preserve the effective edge information of the aliasing region.

In addition, when the reflected grinding surface has a large roughness, the results in Figure 13 illustrate that FCM_S1, FCM_S2, FGFCM and FCM_SNLS has bad segmentation performance. FLIFCM, and KWFLICM obtain wrong segmentation results in some areas due to only the neighborhood spatial information is used. Moreover, FCM_NNS and CKS_FCM achieve satisfactory results for removing the effect of the noises, and the proposed FCM_NNS algorithm are superior to other algorithms for the effective retention of the details in the aliasing region.

Figure 14 gives the *v_pc_* and *v_pe_* values of the proposed and the compared algorithms on thirty aliasing images reflected by grinding samples with different roughness. In Figure 14, with the increase of sample serial number *N*, the surface roughness of grinding samples increases gradually. It can be seen found Figure 14 that the surface roughness affects the segmentation performance. The curve of partition coefficient *v_pc_* reduces with the increase of roughness and the curve of partition entropy *v_pe_* rises with the increase of roughness. In addition, it can be found that FCM_NNS has some certain advantages on validity functions *v_pc_* and *v_pe_*, but the advantages are not obvious. The *v_pc_* value of FCM_NNS is larger than that value obtained by FCM_S1 and FCM_S2, and the *v_pe_* value of FCM_NNS is less than that value obtained by FCM_S1 and FCM_S2.

The idea of using *v_pc_* and *v_pe_* is that the partition with less fuzziness means better performance. However, the effective edge information in aliasing images is highly fuzzy, too ideal *v_pc_* and *v_pe_* will lead to the loss of fuzzy information in aliasing images, but contrary to the accuracy measurement of surface roughness. In Figure 12 and Figure 13, it can be found that misclassified pixels of FLICM, KWFLICM and CKS_FCM incorporating fuzzy factor *G_ij_* are lumped together in transitional region. This is because in order to obtain better performance on *v_pc_* and *v_pe_*, they force the pixels in transitional region to be classified into a certain class with higher membership, which may damage the effective fuzzy information of pixels. Thus, although the fuzzy factor *G_ij_* has strong robustness to noise, it has poor performance in accurately describing the membership of high fuzzy image pixels. Because our ultimate goal is to measure the surface roughness by analyzing the image aliasing degree, and the proposed *F* index is based on the membership information, so these clustering algorithm-based roughness measurements need to be further evaluated by comparing the performance of the roughness index obtained by the corresponding clustering results.

The segmentation results of FCM_NNS on aliasing images reflected by grinding samples with different roughness are shown in Figure 15. These results illustrate that with the surface roughness increases, the area of aliasing region increases monotonically. Therefore, the feature index can be designed based on the clustering results to evaluate the surface roughness. The effectiveness of the proposed *C* index and *F* index are validated by the comparisons between nine roughness indexes in the next section.

Moreover, it can be found from Figure 6 and Figure 15 that when the surface roughness is small, the noise level of aliasing image is low, and the segmentation result is better than other three images in the meanwhile. And when the surface roughness is large, the segmentation results become worse due to the influence of surface textures and other machining marks.

Finally, the time cost of these clustering algorithms on the images with different size is investigated, as shown in Figure 16. All experiments were performed on an Intel(R) Xeon(R) CPU E5-2620 V4 @ 2.10 GHZ, 2.10 GHZ, Windows 10 computer using MATLAB 2017b. It can be seen from Figure 16 that the time cost is similar when the image size is small. With the increasing of the image size, the CKS-FCM and KWFLICM algorithms are more time consuming than other algorithms. The proposed FCM_NNS algorithm is much faster than other compared algorithms.

The results of two groups of experiments indicate that the proposed algorithm performs well. Misclassified pixels are reduced and *SA* is significantly improved compared to other seven algorithms on Synthetic images and the proposed algorithm can not only preserve robustness and noise insensitiveness, but also preserve the effective fuzzy information of aliasing images. Moreover, the computing speed of FCM_NNS is much faster than other compared algorithms, which meets the requirement of in-process measurement of surface roughness in grinding process.

### 4.4. Comparison of Roughness Assessment Indexes

The above clustering results indicate that with the increase of surface roughness, the area of aliasing region increases monotonically. Therefore, the feature index can be designed based on the clustering results to evaluate the surface roughness. Generally speaking, the fuzzy images can be first converted to the crisp segmented images by Equation (28). After that, the surface image can be characterized by counting the number of pixels in the aliasing region class (*C* index). However, the membership information in the fuzzy image is ignored in this index. This has bad effects on the accurate measurement of surface roughness. To fully utilize the effective information in clustering results, another index (*F* index) is designed in Equation (29). The coefficient of determination *R*^2^ values computed by FCM_NNS based on *F* index with different parameter p is shown in Figure 17.

As shown in Figure 17, with the increases of *p* from 0.05 to 0.65, the coefficient of determination *R*^2^ value increases. When *p* is 0.65, the value of *R*^2^ reaches its maximum value. And when p is larger than 0.65, the value of *R*^2^ begins to decrease. The best fitting result is obtained when the value of *R*^2^ is maximal. Hence, the parameters setting are p=0.65.

Figure 18 illustrates the fitting results between the surface roughness and FCM_NNS based indexes on the thirty aliasing images. The number of pixels in the aliasing class (*C* index) versus the surface roughness is shown Figure 18a. Figure 18a indicates that as the roughness increases, the number of pixels in the aliasing region class increases. The regression equation and the *R*^2^ for the relationship between *C* index and roughness are given by:(34)$$y=55103x+52246\;\;\;\;\;\;\;\;\;\left(R^{2}=0.8341\right)$$
where y is the number of pixels in the aliasing region class; x is the surface roughness.

The regression equation and the *R*^2^ for the relationship between *F* index and roughness (Figure 18b) are given by:(35)$$y=0.25x+0.32\;\;\;\left(R^{2}=0.9327\right)$$
where y is the value of *F* index; x is the surface roughness.

From Figure 18, it can be found that the FCM_NNS algorithm can be used in roughness measurement. The experimental results indicated that the FCM_NNS based segmentation results can characterize the aliasing degree in the surface image and highly correlated with roughness. ($R^{2}=0.9327$ for the *F* index and 0.8341 for the *C* index). To prove the superiority of the proposed measurement, the correlation between different indexes and roughness are compared in the following.

The average of the gray values *Ga* [7,20] is the main roughness index in spatial domain, and the major peak frequency *F1* [10,20] is the representative roughness index in frequency domain. The entropy *En* index represents the image’s information content, which can be used to measure the roughness [48,49]. Recently, the color information based indexes have been studied to measure the roughness. The color difference *CD* [7], the overlap degree *S* [21], the mixing region area *MRA* and the relative mixing degree *RMD* [17] are four recently proposed color indexes. Therefore, the indexes used for comparison are *Ga*, *F1*, *En*, *CD*, *S*, *MRA*, *RMD* and the proposed *C* and *F* indexes in Equations (28) and (29).

It can be found from Table 4 that *Ga*, *F1* and *En* have poorer fitting effects than the other four color information indexes, which proves the effectiveness of the color information-based indexes. Then, the results in Figure 18, Table 4 and Table 5 show that the proposed FCM_NNS based *F* index is strongly correlated with surface roughness even if aliasing images are at high noise levels. The value of *R*^2^ for FCM_NNS based *F* index is higher than other compared roughness indexes, which can verify the feasibility and superiority of the proposed surface roughness measurement. The proposed FCM_NNS can not only preserve robustness and noise insensitiveness, but also preserve effective fuzzy details in aliasing images, so the values of *R*^2^ for FCM_NNS-based *C* index and *F* index are larger than that values of other compared clustering algorithms. Moreover, *F* index is designed based on the fuzzy information in the segmentation images, which has better fitting effect than *C* index.

## 5. Conclusions

In this study, a robust FCM algorithm with non-neighborhood spatial information is proposed for surface roughness measurement. The proposed FCM_NNS algorithm can analyze the aliasing degree of a surface image, which can overcome the disadvantages of the FCM algorithms with neighborhood spatial information. Specifically, a method for obtaining appropriate initial cluster centers is proposed firstly to enable the FCM_NNS algorithm converges to the global optimum rapidly. In order to improve the robustness to noise and preserve effective fuzzy details in aliasing images, the non-neighborhood spatial information is extracted from those aliasing images. In the proposed algorithm, the adaptive scale factors λ_1_, λ_2_ are directly determined by the noise level of the aliasing image. The experimental results indicate that FCM_NNS is very effective and efficient.

To fully utilize the fuzzy information in clustering results, the *F* index is designed based on the partition matrix to evaluate the surface roughness. The comparison of roughness assessment indexes indicate that the proposed FCM_NNS based *F* index is strongly correlated with surface roughness even if aliasing images are at high noise levels. The coefficient of determination *R*^2^ is 0.9327 for thirty grinding samples. Moreover, the value of *R*^2^ for FCM_NNS based *F* index is higher than that values of other roughness indexes, which can verify the feasibility and superiority of the proposed surface roughness measurement method.

## Figures and Tables

**Figure 1 sensors-19-03285-f001:**
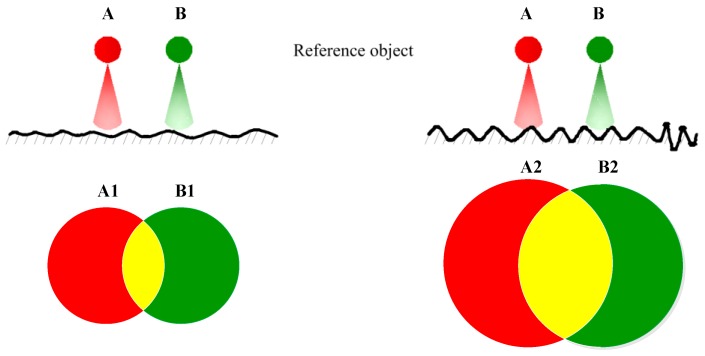
Reference object and its virtual image on different roughness surfaces.

**Figure 2 sensors-19-03285-f002:**
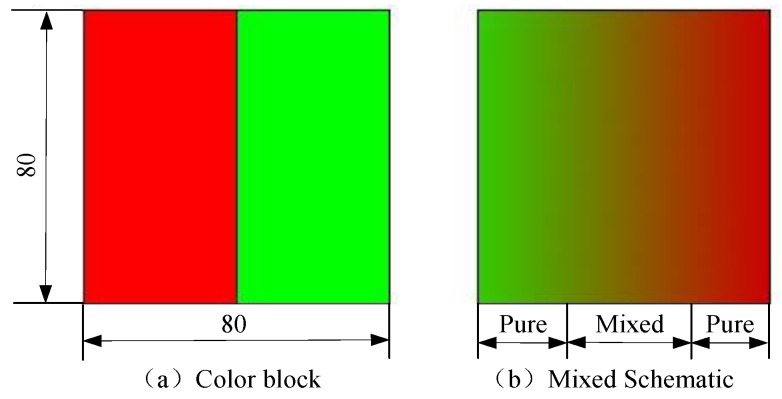
Color block and the schematic of image aliasing.

**Figure 3 sensors-19-03285-f003:**
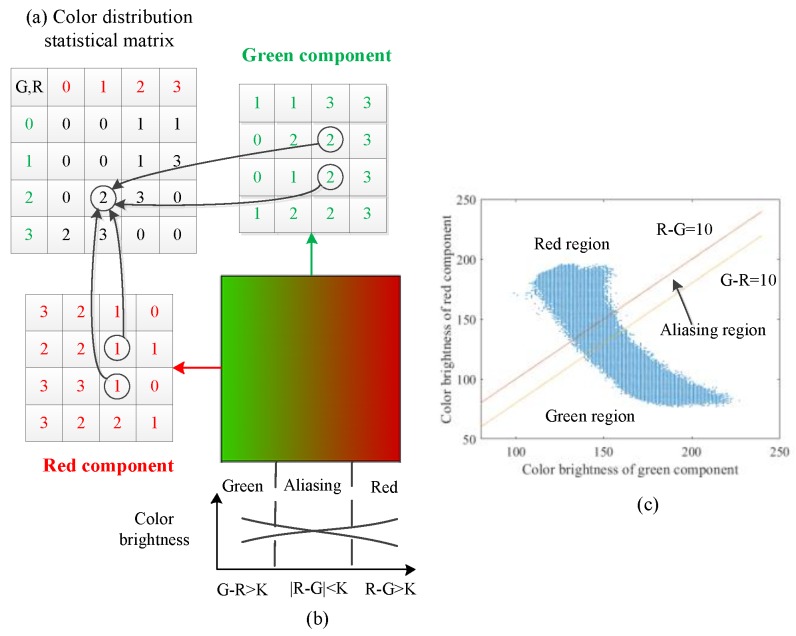
Initial cluster centers based on CDSM.

**Figure 4 sensors-19-03285-f004:**
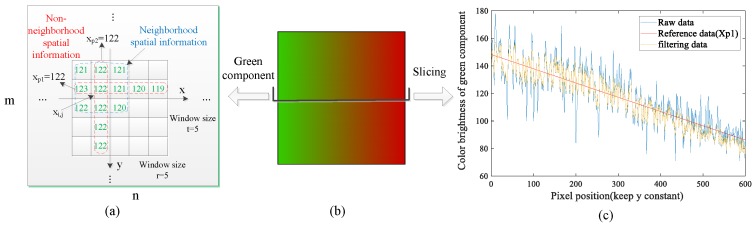
Non-neighborhood spatial information. (**a**) Spatial information, (**b**) Aliasing image, (**c**) Non-neighborhood spatial information in the direction of x.

**Figure 5 sensors-19-03285-f005:**
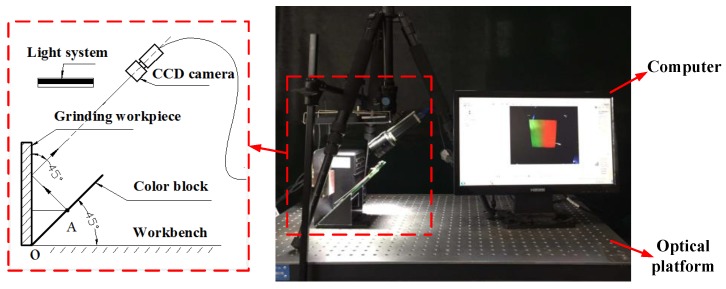
Experimental model and device.

**Figure 6 sensors-19-03285-f006:**
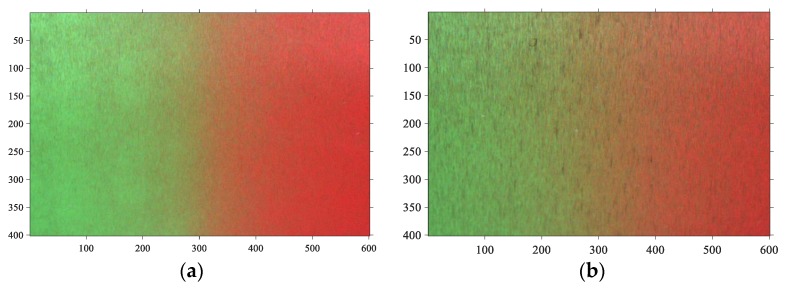
Aliasing images (the roughness is 0.0392, 0.1697, 0.2964, 0.4261 μm, respectively). (**a**) 0.0392 μm, (**b**) 0.1697 μm, (**c**) 0.2964 μm, (**d**) 0.4261 μm.

**Figure 7 sensors-19-03285-f007:**
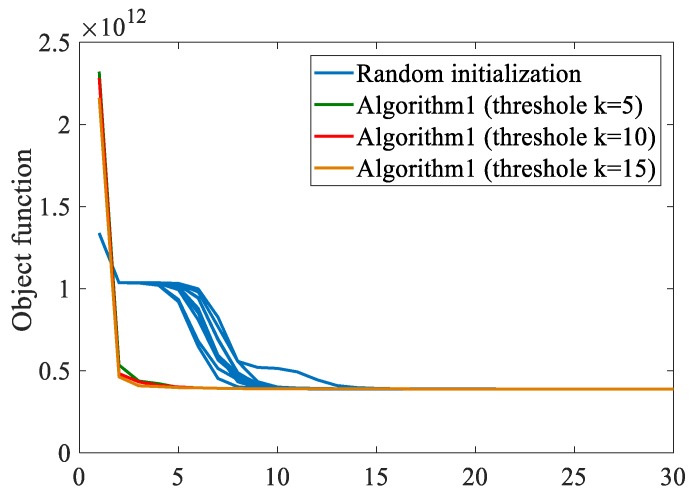
Object function values of FCM_NNS with different number of iterations.

**Figure 8 sensors-19-03285-f008:**
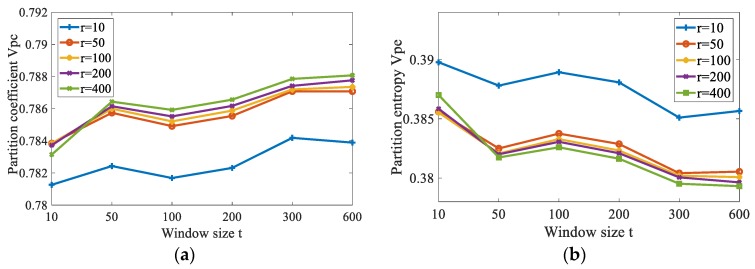
The partition coefficient *v_pc_*, and partition entropy *v_pe_* of FCM_NNS with different size of non-neighborhood information. (**a**) The partition coefficient *v_pc_*, (**b**) The partition entropy *v_pe_*.

**Figure 9 sensors-19-03285-f009:**
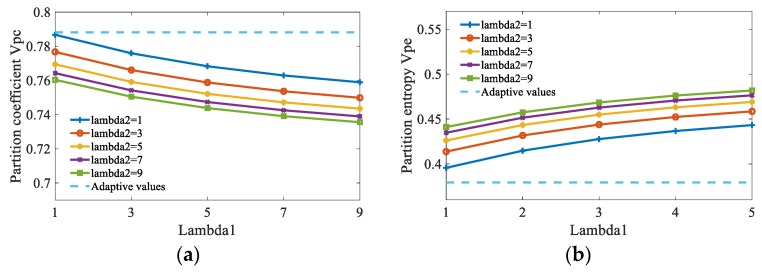
The partition coefficient *v_pc_*, and partition entropy *v_pe_* of FCM_NNS with different scale factors. (**a**) The partition coefficient *v_pc_*, (**b**) The partition entropy *v_pe_*.

**Figure 10 sensors-19-03285-f010:**
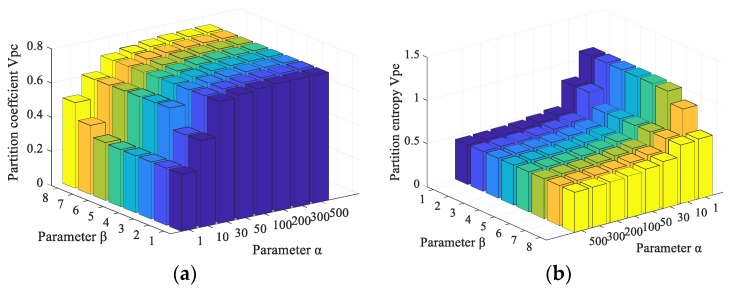
The partition coefficient $v_{pc}$ and partition entropy $v_{pe}$ values of FCM_NNS with different α and β. (**a**) The partition coefficient $v_{pc}$, (**b**) The $v_{pe}$ partition entropy values.

**Figure 11 sensors-19-03285-f011:**
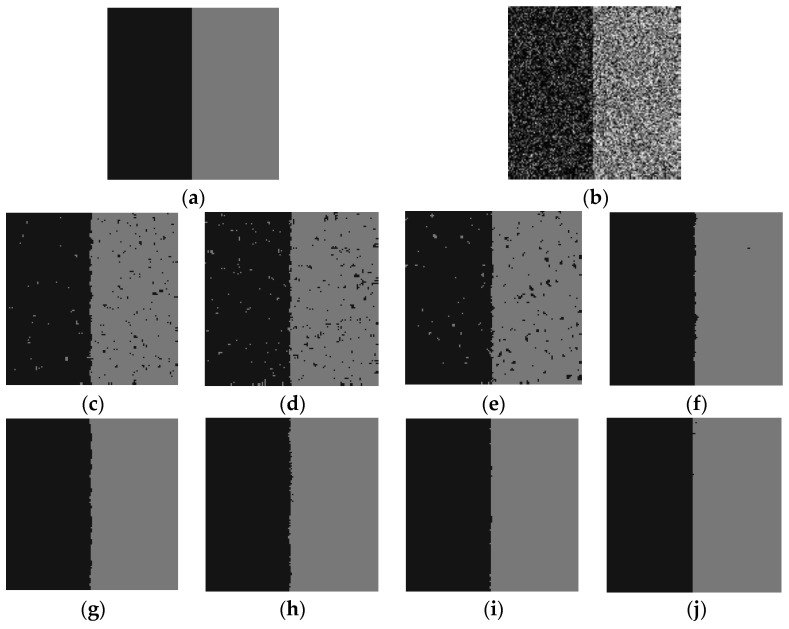
Segmentation results of a synthetic image. (**a**) Original image, (**b**) the same image corrupted by Gaussian noise (20%), (**c**) FCM_S1 result, (**d**) FCM_S2 result, (**e**) FGFCM result, (**f**) FLICM result, (**g**) KWFLICM result, (**h**) FCM_SNLS result, (**i**) CKS_FCM result, (**j**) FCM_NNS result.

**Figure 12 sensors-19-03285-f012:**
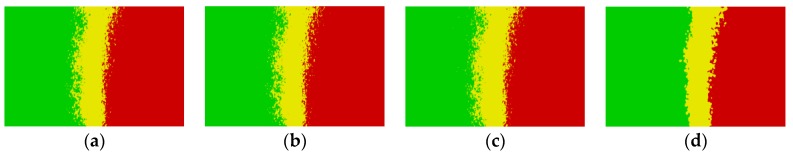
Segmentation results on the first aliasing image which roughness is 0.0392 μm. (**a**) FCM_S1 result, (**b**) FCM_S2 result, (**c**) FGFCM result, (**d**) FLICM result, (**e**) KWFLICM result, (**f**) FCM_SNLS result, (**g**) CKS_FCM result, (**h**) FCM_NNS result.

**Figure 13 sensors-19-03285-f013:**
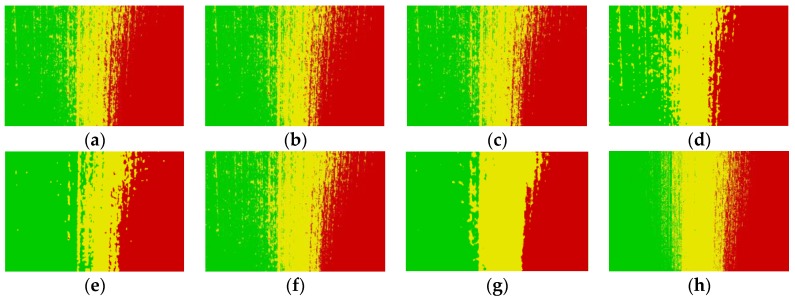
Segmentation results on the second aliasing image which roughness is 0.4261 μm. (**a**) FCM_S1 result, (**b**) FCM_S2 result, (**c**) FGFCM result, (**d**) FLICM result, (**e**) KWFLICM result, (**f**) FCM_SNLS result, (**g**) CKS_FCM result, (**h**) FCM_NNS result.

**Figure 14 sensors-19-03285-f014:**
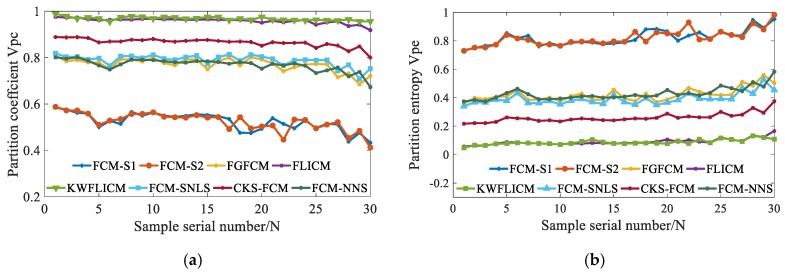
The *v_pc_* and *v_pe_* values of the proposed and the compared algorithms on thirty aliasing images. (**a**) The *v_pc_* values of the proposed and the compared algorithms on thirty aliasing images, (**b**) The *v_pe_* values of the proposed and the compared algorithms on thirty aliasing images.

**Figure 15 sensors-19-03285-f015:**
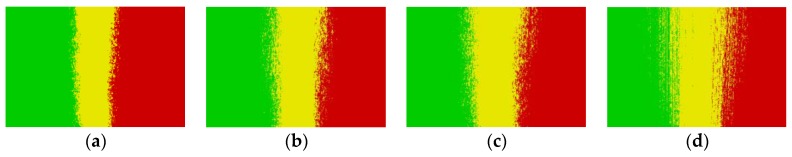
Segmentation results on four aliasing images whose roughness is 0.0392, 0.1690, 0.2964, 0.4261 μm, respectively. (**a**) 0.0392 μm, (**b**) 0.1690 μm, (**c**) 0.2964 μm, (**d**) 0.4261 μm.

**Figure 16 sensors-19-03285-f016:**
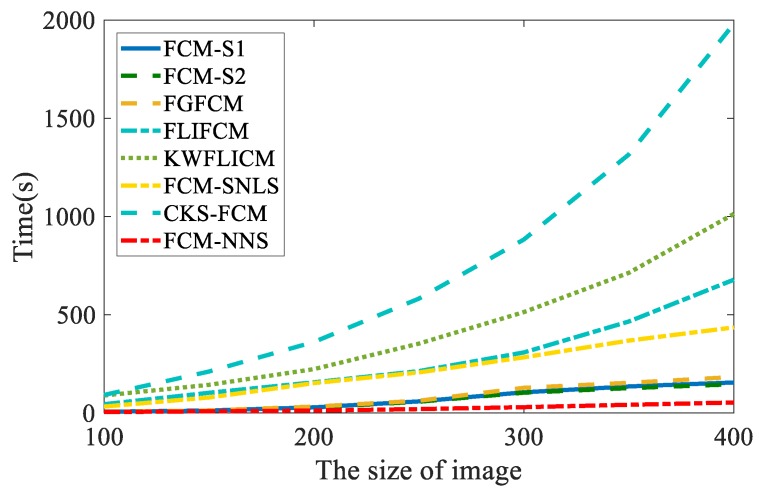
Running time of the eight algorithms.

**Figure 17 sensors-19-03285-f017:**
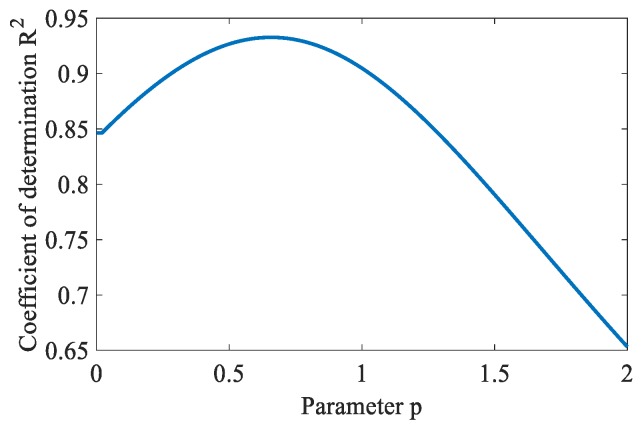
*R*^2^ values of FCM_NNS based *F* index with different parameter *p*.

**Figure 18 sensors-19-03285-f018:**
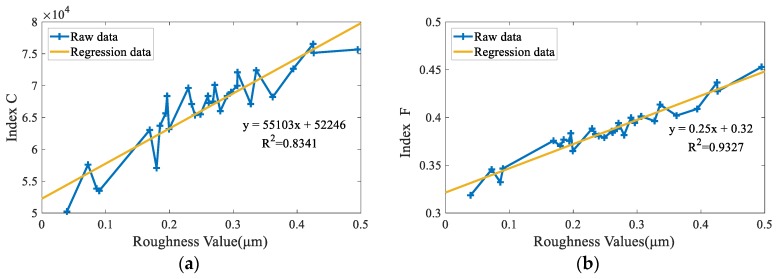
Raw data and regression data. (**a**) FCM_NNS-based *C* index (**b**) FCM_NNS-based *F* index.

**Table 1 sensors-19-03285-t001:** Notations and their descriptions used in the proposed algorithm.

Notation	Description	Notation	Description
K	Threshold in Algorithm 1	R/G	Red/green components
$U=\{u_{ij}\}/V=\{v_{i}\}$	Membership matrix/centers	$x_{p1},x_{p2}$	Reference values
$x_{j}/\bar{x}$	The *j*th pixel/average value of pixel	$a_{i},b_{i}$	Linear regressive coefficient
$x_{i,j},x_{i,j,p1},x_{i,j,p2}$	The pixel in *i*th row, *j*th column and corresponding reference values	t,r	Size of non-neighborhood window
η	Filtered image	$E_{1},E_{2}$	Similarity measure
$\lambda_{1},\lambda_{2}$	Scale factors	$\sigma_{1},\sigma_{2}$	Scale parameters
d	Distance between pixel and it reference value	c/N	The number of centers/the number of pixels
m	Weighting exponent	α,β	Parameters in objective function
C,F	Roughness correlated indexes	$z_{i}$	The membership degree of *i*th pixel belonging to the aliasing region class
$I/I_{s}$	Input image/segmented image	ε/T	Stopping condition/maximum iteration number
$J_{m}/n\left(K\right)$	Objective function/between-cluster separation term parameter	p	Weighting exponent

**Table 2 sensors-19-03285-t002:** Comparative methods.

Method	Input Parameters	Appearance in
FCM_S1 [43]	$\beta ,S_{R}$	IEEE SMC Part B (2004)
FCM_S2 [43]	$\beta ,S_{R}$	IEEE SMC Part B (2004)
FGFCM [38]	$\lambda_{g},\lambda_{s}$	PATTERN RECOGNITION (2007)
FLICM [35]	$S_{R}$	IEEE Image Processing (2010)
KWFLICM [36]	$S_{R}$	IEEE Image Processing (2013)
FCM_SNLS [40]	β,r,s	SIGNAL PROCESSING (2011)
CKS_FCM [46]	$\beta ,S_{R},b$	IEEE J-STARS (2016)

**Table 3 sensors-19-03285-t003:** Segmentation accuracy (*SA*%) on the synthetic image.

Noise	FCM_S1	FCM_S2	FGFCM	FLICM	KWFLICM	FCM_SNLS	CKS_FCM	FCM_NNS
Gaussian 15%	98.29	97.15	97.60	99.76	99.80	99.70	99.73	99.96
Gaussian 20%	94.65	92.63	93.80	99.37	99.42	99.59	99.68	99.93
Gaussian 30%	85.17	83.56	84.13	95.67	96.53	99.32	99.12	99.87
Salt & Pepper 15%	99.10	99.89	98.60	99.92	99.93	99.93	99.94	99.96
Salt & Pepper 20%	97.35	99.72	95.71	99.88	99.92	99.81	99.87	99.95
Salt & Pepper 30%	83.52	93.71	84.12	99.52	99.63	98.36	99.32	99.93

**Table 4 sensors-19-03285-t004:** Correlation coefficient values of comparative indexes.

	*Ga*	*F1*	*En*	*S*	*CD*	*MRA*	*RMD*
*R* ^2^	0.7179	0.7261	0.7174	0.7786	0.7916	0.8173	0.7984

**Table 5 sensors-19-03285-t005:** Correlation coefficient values of the improved FCM-based roughness indexes.

	FCM_S1	FCM_S2	FGFCM	FLICM	KWFLICM	FCM_SNLS	CKS_FCM	FCM_NNS
*R*^2^ (Index *C*)	0.2755	0.1556	0.6781	0.7935	0.7673	0.5800	0.7834	0.8341
*R*^2^ (Index *F*)	0.8752	0.8712	0.7402	0.8149	0.8342	0.7061	0.8981	0.9327

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
