# Peer review of "Fuzzy Clustering Algorithm with Non-Neighborhood Spatial Information for Surface Roughness Measurement Based on the Reflected Aliasing Images"

_sensors, 2019, doi:10.3390/s19153285_

Round 1

Reviewer 1 Report

Following recommendations must be followed:

In the eq. (3): What happen if m=1?, which are the initial value conditions for "m"?

In general in the whole paper, but for example: in the line 144 - which are the values of epsilon and T?, it must be clarified that all of the parameter values suggested are defined in their magnitude after.

In the line 127: What is the meaning of EnFCM?.

In the line 138 and 139: What does mean RFLICM and KWFLICM?

In line 149: where is the non-local neighborhood information present in the algorithm, is not clear justified, and there are a lot of varibales undefined such as N_{j}, N_{p}.

In the Fig. 3b): G-R > K and R-G > K, must be absolute difference values?

A better explanation of the Fig. 3c): is not understandable.

In general, the Title of the paper must be modified, the proposal is applied for specific types of images or, the same quality results happen for real color images or more complex images?

In the eqs (12) and (13): I think there is a mistake, the coeficient "k" can not be used as magnitude value in both equations because it dictates a parameter that goes from [1,...., r or t] and does not take a magnitude value from the non-neighborhood window processing. CHECK THAT!

In the line 244: which is the role of lambda_{1} and  lambda_{2}?

In the eq. (25), as I said before, must be defined Beta, and other variables named first of their definition.

In the eq. (28): sub-index "j" goes form ... through ...?

In the line 305: "u" variable is defined as in the eq. (26)?

In the line 306, ""p" is similar to the fuzzification factor "m"" , but, is computed in the same way?

Again, in the Algorithm 2 in the point 1: There are a lot of parameter values to be considered, perhaps is better to include a Table including all the parameters proposed to sintonize your algorthm and their respective rank values!

Take care the style of the paper, in the point 8 of the algorithm 2, the last prhase "go the 6" is better if it says, "go to the 6th step"

In line 316: To improve the style, please add REFS. for every one of the algorithms in this section.

In the Fig 6: "x" and "y" indexes are in pixels?, Which is the resolution of your system?
Which are the particular characteristics of the images captured from your system?. In the caption: The roughness dimensionality is totally obtained from the capturing device?

Reviewer 2 Report

The basic contribution of this paper is to replace the neighborhood spatial information by a non-neighborhood spatial information for FCM-based image processing of a two-color aliasing image. The non-neighborhood window of a pixel is defined as the row and column pixels as opposed to the neighborhood window which is comprised of the closest adjoining pixels. There is no justification provided in the paper except by the way of results that the non-neighborhood window will have more pertinent information than the neighborhood window will. The empirical results tend it show it but it could be contingent on the images that you selected for the analysis and may not work for other surface roughness images. 

Also explain how would one chose x and y directions in the image if the machining process and underlying surface roughness profile is apriori unknown. For example, if an end mill was fed in one direction the surface roughness profile would be very different and therefore the image and the aliasing direction compared to if the same end mill was fed in the perpendicular direction. 

The writing needs to be improved although overall the paper is well structured.     
